# Quantitative estimates of average geomagnetic axial dipole dominance in deep geological time

Andrew J. Biggin [1✉], Richard K. Bono[1], Domenico G. Meduri[1], Courtney J. Sprain[1,2], Christopher J. Davies [3], Richard Holme[1] & Pavel V. Doubrovine[4]

A defining characteristic of the recent geomagnetic field is its dominant axial dipole which provides its navigational utility and dictates the shape of the magnetosphere. Going back through time, much less is known about the degree of axial dipole dominance. Here we use a substantial and diverse set of 3D numerical dynamo simulations and recent observation-based field models to derive a power law relationship between the angular dispersion of virtual geomagnetic poles at the equator and the median axial dipole dominance measured at Earth's surface. Applying this relation to published estimates of equatorial angular dispersion implies that geomagnetic axial dipole dominance averaged over $10^7$–$10^9$ years has remained moderately high and stable through large parts of geological time. This provides an observational constraint to future studies of the geodynamo and palaeomagnetosphere. It also provides some reassurance as to the reliability of palaeogeographical reconstructions provided by palaeomagnetism.

[1] Department of Earth, Ocean and Ecological Sciences, University of Liverpool, Liverpool L69 7ZE, UK. [2] Department of Geological Sciences, University of Florida, PO Box 112120, Gainesville, FL 32611-2120, USA. [3] School of Earth and Environment, University of Leeds, Leeds LS2 9JT, UK. [4] Centre for Earth Evolution and Dynamics, University of Oslo, Sem Saelands vei 2A, 0315 Oslo, Norway. ✉email: biggin@liverpool.ac.uk

A primary feature of the geomagnetic field today is its strong axial dipole (AD) component which provides an effective shield against the solar wind[1,2] helping to make the planet habitable[3]. The field is highly variable in time however and our knowledge of its morphology declines rapidly as we go back in geological history. At a given instance, the degree of AD dominance over the remaining non-axial dipole (NAD) components at Earth's surface may be expressed, here, in terms of the Lowes power[4] for the magnetic field energy ($W$) as[5]

$$\mathrm{AD/NAD} = W_1^0/(W - W_1^0),\qquad(1)$$

where

$$W = \sum_{n=1}^{n_{max}}\sum_{m=0}^{n} W_n^m\qquad(2)$$

and

$$W_n^m = (n+1)\left[(g_n^m)^2 + (h_n^m)^2\right].\qquad(3)$$

Here $g_n^m$ and $h_n^m$ are the Gauss coefficients of degree $n$ and order $m$ for the spherical harmonic expansion of the geomagnetic potential[6]; $g_1^0$ is the AD component.

The current geomagnetic field has AD/NAD of ~10 (Fig. 1) but, according to time-dependent global magnetic field models[7–11], this has varied by more than one order of magnitude on timescales of kyr (Supplementary Fig. 1) over the last 100 kyr. By definition, AD/NAD must briefly fall to zero during a polarity reversal and

can also fall far below unity during excursions[7]. To avoid biasing by brief extreme events, we will take the median of the instantaneous AD/NAD ratios which we call AD/NAD$_{median}$ as our measure of average AD dominance. We note that this value is a first-order description of the average, time-instantaneous field morphology and is not intended as a direct measure of the validity of the geocentric axial dipole (GAD[12]) hypothesis, which rather would rely on the morphology of the time-average field (TAF). The TAF is defined by time-averaging all Gauss coefficients independently before using their ratios to define its properties, which may be very different to the properties of the instantaneous field at any and all times. For example AD/NAD$_{TAF}$ is, by definition, infinite for a GAD field whereas the associated AD/NAD$_{median}$ value may be finite and even small. In this sense, AD/NAD$_{median}$ is more relevant to those using palaeomagnetic records to understand geomagnetic behaviour, core dynamics and the magnetospheric shielding it confers than to those interested in making tectonic reconstructions. The implications of this study for palaeogeographical reconstructions is nevertheless explored later.

Direct estimates of AD/NAD$_{median}$ and other useful ratios are possible from statistical field models based on the Giant Gaussian Process[13–17] spanning back to 10 Ma (Supplementary Table 1). Previous efforts to assess the average morphology of the palaeomagnetic field prior to 10 Ma have been forced to rely on the Model G approach[18] to analysing palaeomagnetic secular variation data. This relies on measurements of the angular dispersion ($S$; see the "Methods" section) of virtual geomagnetic poles

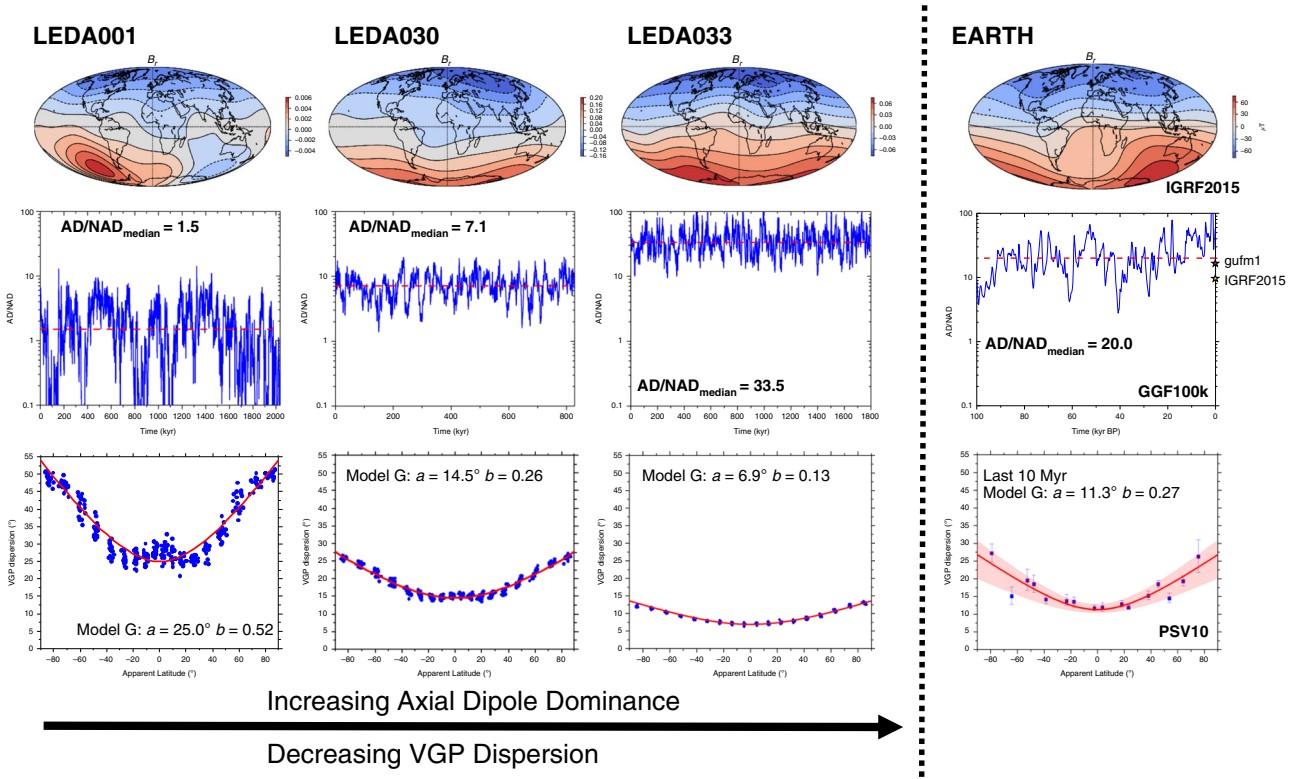

**Fig. 1 Summary of magnetic field behaviour from representative geodynamo simulations and Earth.** The dynamo simulations (first three columns) show the tendency of AD/NAD$_{median}$ to increase as VGP dispersion decreases. The first row is a snapshot of radial field at the Earth's surface taken from a timestep with AD/NAD close to its median for the time series shown in the second row (median shown as red line; note the semi-log scale). The third row represents palaeosecular variation as presented in studies of ancient time periods (but with far more data); the red line represents the best Model G fit (parameters $a$ and $b$ provided) to the entire data set of 500 randomly drawn timesteps (blue circles) sampled at each of 324 regularly placed locations (see the "Methods" section). Final column: equivalent plots for observational geomagnetic models[11,63] and a palaeosecular variation dataset from last 10 Myr[19]. Note that AD/NAD for IGRF2015 is 9.6 and AD/NAD$_{median}$ for gufm1 is 16.4. The red shaded area around the Model G fit to the empirical data represents 95% confidence bounds.

(VGPs) recovered from collections of palaeomagnetic recorders (normally lavas). Model G has the form of a second-order polynomial:

$$S^2 = a^2 + (b\lambda)^2, \qquad (4)$$

where $a$ and $b$ are constants that define the value of $S$ at the equator and the rate of its increase with palaeolatitude ($\lambda$), respectively. Using PSV10, a recent compilation of palaeomagnetic secular variation data from rocks formed within the last 10 Myr[19], these Model G constants, given together with their 95% confidence limits, were recently calculated[20] as $a = 11.3^{+1.3°}_{-1.1}$ and $b = 0.27^{+0.04}_{-0.08}$. For older datasets, the palaeolatitude must be estimated using the palaeomagnetic data themselves and this approach was simulated here (see "Methods" section).

Using insights from mean-field kinematic dynamo theory and the modern field, McFadden et al.[18] made the case that Model G could be used to represent the relative importance of two independent dynamo "families". The constant $a$ denoted the magnitude of the secular variation in the "quadrupole family" comprising those spherical harmonic terms which are symmetric with respect to the equator (and include the equatorial dipole terms). Likewise, $b$ did the same for anti-symmetric terms (including the AD) comprising the "dipole family". In the context of this approach, intervals of time whereby the AD and related antisymmetric terms were particularly dominant over the symmetric terms should be recognisable through increased values of $b$ relative to $a$ in Model G fits to PSV datasets. Such intervals have previously been argued to include the Cretaceous Normal Superchron[21–23] and much of Precambrian time[5,24–27] but these claims are difficult to verify since the premise on which Model G fits are interpreted is oversimplified[28].

Here we develop and apply a more robust approach to ascertaining information regarding the morphology of the ancient geomagnetic field using palaeosecular variation data. The power law that we derive is used to obtain quantitative estimates of ancient AD dominance showing that this has been maintained at near-present-day values for much of Earth history.

## Results and discussion

**Model G relationships from dynamo simulations.** For the purposes of this study, we use the outputs of 61 numerical dynamo simulations (see "Methods" section; Supplementary Data 1) which were required to be run for a sufficient amount of time (>100 kyr) to obtain a reasonable temporal sampling of the simulated magnetic field behaviour at the Earth's surface. Each model was distinct in terms of its input parameters and diverse physical ingredients were represented. These included homogeneous and heterogeneous outer boundary heat flux conditions and small and present-day inner core sizes. Models with internal heating sources derived from radiogenic heating, with a stably stratified layer at the top of the core, as well as models where convection is purely chemically driven were also employed (see "Methods" section). The resulting field behaviour ranges from exhibiting $S$ and AD/NAD$_{median}$ values much greater than the Earth's values for recent times to much lower values (Fig. 1). In most cases, Model G (after applying a variable cutoff[29] for outliers in VGP distributions) provided a good, though not perfect, fit to VGP dispersion data across the apparent latitudes (Supplementary Data 1) yielding root mean square error (RMSE) values with a median across all models of 1.2°. Model G $a$ and $b$ parameters are positively correlated as are the powers of the Gauss coefficients (except $g_1^0$) with degree and order that sum to odd values ($W_{ODD}$) and even values ($W_{EVEN}$) (Supplementary Fig. 2). As VGP dispersion increases, so does its latitudinal dependence and this reflects increases in the nonaxial-dipole field being

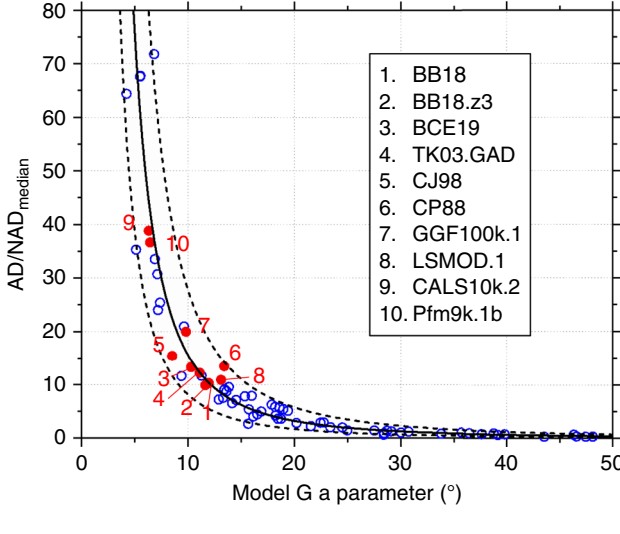

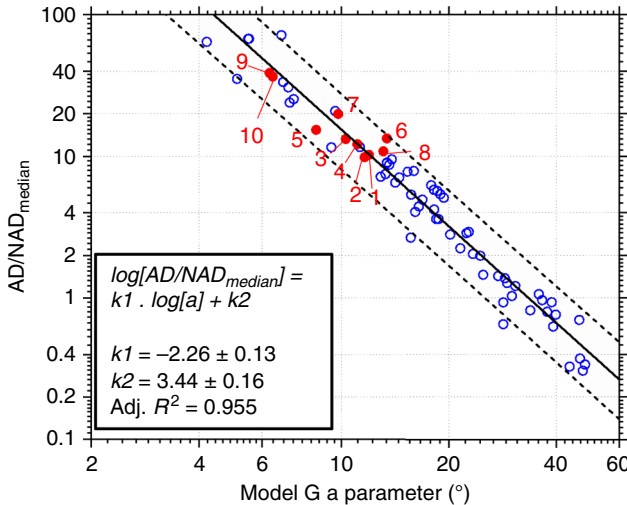

**Fig. 2 Power-law relationship, shown on linear and log axes, enabling estimation of first-order geomagnetic field morphology from palaeosecular variation analysis.** Red points are observation-based models (Supplementary Table 1) testing the relationship which is based entirely on dynamo simulation outputs (blue hollow points; Supplementary Data 1). Shaded area is 95% prediction bounds calculated from a linear regression performed in log-space.

partitioned similarly into antisymmetric (given by $W_{ODD}$) and symmetric (given by $W_{EVEN}$) terms. We also note in passing that less dipolar simulations in particular tended to produce more complicated curves with an equatorial peak in VGP dispersion (see e.g. LEDA001 in Fig. 1). This implies that a reasonable latitudinal distribution of observations is required to obtain both Model G parameters to a good degree of accuracy.

Having ascertained that the structure of Model G (Eq. (4)) provides efficient two-parameter descriptions for a wide range of simulated PSV behaviour, we explored the potential of these simple quadratic fits to predict average morphological characteristics of the generated fields defined as both the median instantaneous and the TAF (Supplementary Fig. 3). The most striking observation is a strong power-law relation between Model G $a$ parameter (average VGP dispersion at the equator; Eq. (4)) and AD/NAD$_{median}$ (Fig. 2) that in log–log space reads

$$\log(\text{AD/NAD}_{median}) = k_1 \log a + k_2, \qquad (5)$$

where the constants and their 95% confidence limits were obtained from standard linear regression: $k_1 = -2.26 \pm 0.13$; $k_2 =$

3.44 ± 0.16. We also observe the following: (1) since Model G $a$ and $b$ parameters co-vary (Supplementary Fig. 2a), the latter is also correlated with AD/NAD$_{median}$ but here the relationship is not quite so strong (Supplementary Fig. 3c); (2) the relatively weak relationship between $b/a$ and $O/E$ implies that the original morphological interpretation of Model G parameters in terms of independent families of equatorially symmetric and antisymmetric spherical harmonic terms[5,18,22] is only moderately supported by our dynamo simulations (Supplementary Fig. 3a, d); (3) intuitively, Model G parameters provide much stronger constraints on the average instantaneous field morphology (Supplementary Fig. 3a–c) than the morphology of the time-averaged field (Supplementary Fig. 3d–f).

The power law (5) presents a potentially powerful new tool linking geomagnetic secular variation and morphology. While the broad observation that enhancing AD dominance suppresses VGP dispersion may be considered intuitive[28], the correlation and significance of the power law (Adj. $R^2 = 0.955$, $P < 10^{-5}$, number of data, $N = 61$, Spearman rank coefficient $\rho = 0.971$) is remarkably and unexpectedly high. We note that a power-law relationship with similar parameters may also be predicted from simple theoretical arguments (see Supplementary Note 1).

**Testing the correlation using observation-based field models.** To ascertain whether time-varying and statistical field models of PSV derived from palaeo-magnetic and geo-magnetic observations yield estimates of AD/NAD$_{median}$ and Model G $a$ values which are consistent with the power law in Fig. 2, we apply the same analytical approach (see "Methods" section) to a selection of these (Supplementary Table 1)[7,8,10,11,13–15]. Although similarly represented by sets of Gauss coefficients, the methods of generating these field descriptions are fundamentally distinct to those used to obtain the outputs of the dynamo simulations. While dynamo simulations model field behaviour by numerically solving equations governing the outer core magnetohydrodynamic processes responsible for it, observational models are defined by fitting spatially and temporally restricted datasets of palaeomagnetic, archaeomagnetic and geomagnetic measurements and their associated age estimates. Another important difference is that three of these observational models are restricted to intervals of 9–20 kyr which may be too short to capture time-averaged field behaviour[30]. The statistical models, on the other hand, assume that the statistical properties of palaeosecular variation can be modelled by a "Giant Gaussian Process", whereby the Gauss coefficients are randomly drawn from normal distributions with means and variances set to produce the desired characteristics of palaeosecular variation and the time-averaged field[13–17] (all models assume independently distributed Gauss coefficients except those of ref. [16], which assumes a covariance among a select set of Gauss coefficients).

Given the above and the varied AD/NAD$_{median}$ values produced by these observational models (red circles in Fig. 2) it is remarkable to observe the Model G parameters are all found close to the power law derived from the dynamo simulations. Indeed, they all fall within an interval (dashed lines) where 95% of future models are predicted to fall according to a t-distribution (see "Methods" section). We note that we are not overly concerned here with the relative realism of any of the outputs shown by these models, merely the ability of their output palaeosecular variation to predict their average morphology.

The robust nature of this geometric relationship is also supported by the results of an analysis summarised in Fig. 3a and Supplementary Fig. 3. Here, the $g_1^0$ term produced at each timestep (or realisation) from three dynamo simulations and one Giant Gaussian Process was rescaled to produce values of AD/NAD$_{median}$ that were radically different from that which the model originally produced (see Supplementary Note 2 for more details). Doing so simultaneously affected the angular dispersion of VGPs such that the resulting $a$ parameter of the Model G fit fell within the prediction bounds of the earlier derived power law (Fig. 3a). This demonstrates that, so long as the power spectrum of the nondipole field is consistent with any of these models, the relationship is robust to a large range of AD/NAD$_{median}$ values.

Based on the evidence presented in Figs. 2 and 3a, Eq. (5) and its associated prediction bounds appear consistent with all available empirical and synthetic datasets. We therefore consider it to provide a robust description of the relationship between geomagnetic variability and morphology allowing reliable estimates of one to act as a proxy for the other.

**Estimating the ancient geomagnetic AD dominance.** Figure 3b, c and Supplementary Fig. 4 demonstrate two further useful properties of the power-law relation outlined above. Firstly, although AD/NAD$_{median}$ may change significantly for sub-intervals within a single model time series, the associated Model G $a$ parameter from the same subinterval also shifts according to the power law. This implies that selections of palaeomagnetic datasets from any interval duration may be useful for estimating the average AD dominance for that same interval. A caveat is that the interval must be sufficiently long such that significant serial correlation of VGP positions is avoided. Based on our sliding window analysis of both observational models and dynamo simulations (Fig. 3b, Supplementary Fig. 4), 50–100 kyr appears to be sufficient for this purpose; this duration is similar to earlier estimates of the time necessary to sample the time-averaged field[30].

The second useful property of the power-law relation is that it remains capable of accurately estimating AD/NAD$_{median}$ even when the number of locations and time steps used to construct the Model G curve are reduced to values that are well within the bounds of palaeomagnetic datasets available for ancient intervals. Table 1 presents a selection of recent published estimates of Model G $a$ parameters for intervals extending back into the Archaean[19,20,27]. The smallest number of locations comprising any single one of these datasets is 19, while the smallest median number of sampling sites per location (representative of time steps) is 15. These values were used as conservative inputs for the downsampling of models (see "Methods" section) whose results are summarised in Fig. 3c (see also Supplementary Fig. 4). So long as the interval is sufficiently long (>50 kyr), the estimates of AD/NAD$_{median}$ were found to be nearly always reliable (accurate, if not necessarily precise).

Figure 4a presents estimates of AD/NAD$_{median}$ calculated using the Model G $a$ parameter for the five studied intervals listed in Table 1. In each case, the Model G parameters were taken directly from the publications and required application of the Vandamme cutoff[29] as used for all models here. These intervals are far longer than the time spanned by any one of the individual estimates of VGP angular dispersion from which the Model G fits were constructed. Furthermore, in the earlier intervals in particular, there are large gaps in the age distribution of the rocks used to obtain the estimate. Therefore, values of AD/NAD$_{median}$ cited for each period should be considered as weighted towards sub-intervals with denser data coverage (Fig. 4a) and may not be representative of sub-intervals (of which 600–1100 Ma is the most striking) where no or very little data currently exists. A further point to note is that AD/NAD$_{median}$ values will also be more heavily influenced by those rock units with low apparent palaeolatitudes since they exert more influence on the $a$ parameter of the Model G fit.

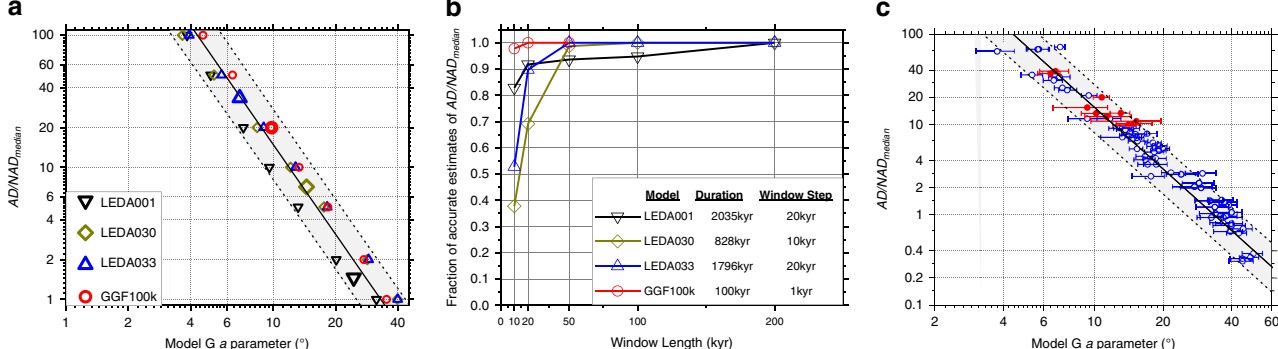

**Fig. 3 Tests of robustness and usefulness of the power law displayed in Fig. 2. a** Effects of arbitrarily rescaling the axial dipole term at every realisation using four models shown in Fig. 1 (see Supplementary Note 2 for details and Supplementary Fig. 3 for individual Model G fits). Original model outputs (large symbols) are diverse; rescaled models (small symbols) can have entirely different values of AD/NAD$_{median}$ to the original but the corresponding Model G fit adjusts simultaneously such that each point remains within the 95% prediction bounds derived from Fig. 2. **b** Results of a sliding window analysis using time series from the same four models as in Fig. 1. In each case, the sliding window of length given by the x-axis was moved from the start of the time series through to the end in window steps of 1−20 kyr (adjusted for the total time series length) drawing 15 random timesteps from each of 19 random locations on the surface of the Earth. Estimates of AD/NAD$_{median}$ made using the power law in Fig. 2 were defined as accurate if they were within calculated uncertainties of the actual value of AD/NAD$_{median}$ for that specific time window. See Supplementary Fig. 5 for individual plots of the AD/NAD$_{median}$ time series and estimates from windows sliding along it. **c** Effects of down-sampling (15 timesteps at each of 19 locations) on 61 dynamo models and 12 observational models on their adherence to the power law shown in Fig. 2. Model G a parameters were calculated from the downsampled dataset whereas AD/NAD$_{median}$ values were calculated directly from the models using every timestep. The overlaps of the majority of calculated uncertainties on Model G a parameters with the prediction bounds ascertained from Fig. 2 indicates that this method of estimating AD/NAD$_{median}$ values from the Model G fit is reliable for palaeomagnetically feasible datasets.

**Table 1 Summary of published palaeosecular variation compilations and associated predictions.**

| Time period | Ref. | $N_{locations}$ | Median $N_{sites}$ | Model G a parameter (°) | Estimated AD/NAD$_{median}$ |
|---|---|---|---|---|---|
| 0–10 Ma | 19,20 | 16[a] | 119 | 11.3 + 1.3/−1.1 | 11.3 + 15.0/−6.5 |
| 84–126 Ma | 20 | 19 | 24 | 10.7 + 2.2/−2.4 | 12.8 + 29.4/−8.3 |
| 127–198 Ma | 20 | 20 | 15 | 12.7 + 1.9/−2.7 | 8.7 + 18.9/−5.3 |
| 0.5–1.5 Ga | 27 | 28 | 17 | 10.1 ± 0.5 | 14.7 + 16.1/−7.6 |
| 1.5–2.9 Ga | 27 | 27 | 17 | 9.2 ± 1.1 | 18.0 + 27.0/−10.5 |

The Model G a parameter for various time periods allows estimation of AD/NAD$_{median}$ from the power law shown in Fig. 2. $N_{locations}$ refers to the number of locations, where S was measured using $N_{sites}$ site-mean palaeomagnetic directions. Uncertainties are reported 95% confidence limits.
[a]In this study, globally distributed VGPs were grouped into 16 latitudinal bins for the purpose of fitting Model G.

The above caveats notwithstanding, the degree of stationarity displayed by our obtained estimates of AD/NAD$_{median}$ is remarkable (Fig. 4a). Uncertainty limits on AD/NAD$_{median}$, calculated by combining uncertainties associated with the Model G a parameters with the power-law prediction bounds, render each time interval indistinguishable from the rest. Furthermore, the total range observed in estimated AD/NAD$_{median}$ values (including 95% uncertainty limits) from 3.5 to 45.0 encompasses the values derived from observation-based field models covering intervals in more recent geological time (Fig. 2).

While AD dominance apparently changes rapidly on short timescales (Supplementary Fig. 1) and is prone to collapse during geomagnetic excursions and reversal transitions, we presently find no evidence that its average over $10^7$–$10^9$ year timescales is subject to significant variations. Given that AD/NAD must instantaneously reach zero for a reversal to take place, a particularly surprising insight is that intervals with substantially different reversal frequencies (e.g. the last 10 Myr, the Cretaceous Normal Superchron, and the early Cretaceous–Jurassic) apparently yield nearly identical values of AD/NAD$_{median}$ (Fig. 4a). This implies that, regardless of how frequently AD/NAD undergoes brief collapse, the field recovers to spend most of its time in a similarly dipole dominated state. Intervals of stable average AD dominance also apparently coincided with significant variations in long-term average field intensity[31–33]. This further

suggests that the magnitude of the AD and NAD field are correlated on long-timescales such that the degree of AD dominance remains approximately constant. These coupled observations may be used as constraints for future geodynamo modelling studies seeking to capture long-term variations in geomagnetic field behaviour.

Changes in aspects of geodynamo behaviour are thought likely to result from secular changes in core cooling modulated by mantle convection over the last several billion years[33–39]. Indeed the changing nature of the forcing of outer core convection from both above and below implies that it is already a challenge to explain how the geomagnetic field has been continuously sustained over Earth history[40]. Here we add the further constraint that models should produce a similar average geomagnetic field morphology for much of a time period where the Earth has seen its liquid core nucleate and grow an inner core[41] and the mantle undergo several supercontinent cycles[42] with consequences expected for core-mantle heat flow and its pattern[43].

Almost all of the numerical dynamo simulations performed in a study[44] aiming to elucidate the magnetic signature of inner core nucleation gave values of AD dominance within the range implied by the palaeomagnetic datasets used here. This suggests that diverse core geometries, control parameters, forcing conditions, etc., are capable of giving rise to field morphologies similar to those associated with Earth in the past. Nevertheless, it is

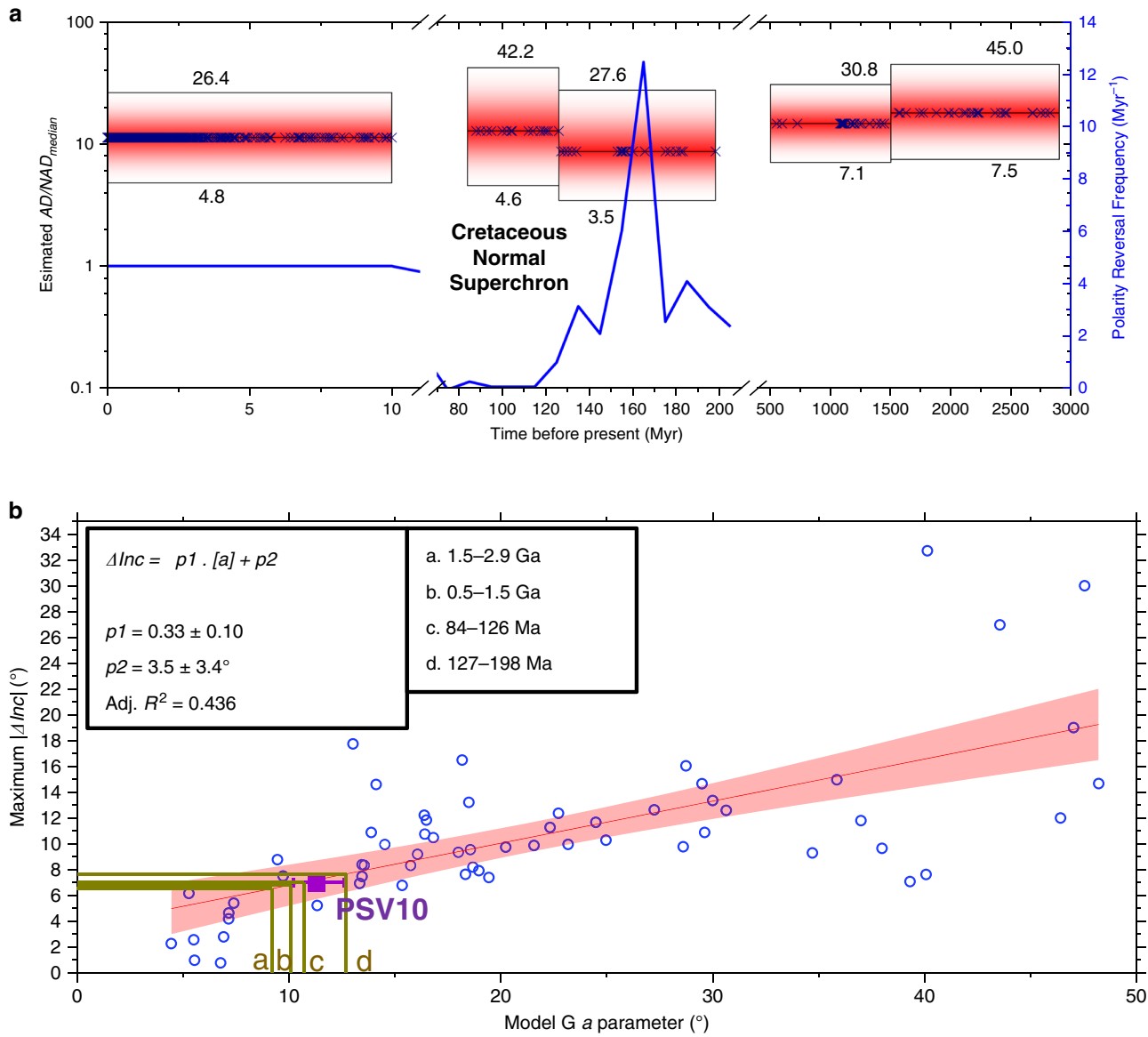

**Fig. 4 Estimates of axial dipole dominance for the ancient geomagnetic field. a** Application of the power law in Fig. 2 to ascertain the first quantitative estimates of axial dipole dominance in deep time (see Table 1). Horizontal range of boxes indicates nominal time range; vertical range indicates 95% uncertainties with numerical bounds provided. Crosses relate to age of one or more rock units comprising the estimate within the box. Reversal frequency was calculated using 10 Myr bins[38]. **b** Relationship between palaeosecular variation and time-averaged inclination anomaly in outputs of dynamo models (blue circles). Shaded area represents 95% confidence bounds. Dataset from the last 10 Myr[19] (purple square) is shown to fit the linear trend well. Extrapolations of inclination anomalies (ΔInc) are made using median *a* parameters for four earlier datasets shown in panel **a**.

important to highlight that our analysis of palaeomagnetic datasets does not rule out exotic field morphologies (e.g. extreme multipolar or equatorial dipole dominated[45]) existing for some times in the past. These could be missed either because of insufficient palaeomagnetic data coverage (Fig. 4) or because their behaviour (and especially their power spectra) is outside the range of models used to constrain the power-law tested here.

Our findings also have implications for Earth's palaeo-magnetosphere and the long-term shielding of Earth's atmosphere from solar wind. The strong and dominantly axial dipolar morphology of the present-day geomagnetic field is an efficient one for reducing fluxes of energetic particles into Earth's upper and middle atmosphere and restricting these to high latitudes[46]. Large reductions in AD dominance, even while maintaining the same dipole moment (e.g. in a pure dipole rotation scenario) are expected to cause polar caps, auroral zones and atmospheric

impacts of solar energetic particles to migrate to lower latitudes[1,47]. For the time periods considered, our results suggest that such major decreases in AD dominance are relatively rare, being restricted to the extremes of reversals and excursions.

A primary application of palaeomagnetism is to produce palaeogeographic reconstructions, making use of the GAD assumption to relate changes in mean inclination to inferred shifts in palaeolatitude. Values of $AD/NAD_{median}$ cannot be interpreted directly as measures of the validity of the GAD approximation of the time-averaged field because they are constructed using different averaging processes (specifically, the former is the average of multiple instantaneous global field morphologies, whereas the latter is the field produced by the average of multiple directional measurements, i.e. the time-averaged power spectrum, and yields $AD/NAD_{TAF}$ values in these models of approximately one order of magnitude higher).

Nevertheless, our dynamo simulations do show correlations between their Model G parameters and AD/NAD$_{TAF}$ (Supplementary Fig. 3) and, most usefully, exhibit a statistically significant relationship between the Model G $a$ parameter, used here to estimate the AD/NAD$_{median}$ values, and the maximum absolute inclination anomaly, a direct and commonly used (e.g. ref. [18]) measurement of the validity of GAD (Fig. 4b; see "Methods" section). Furthermore, our actual measurements of these two parameters using rocks from the last 10 million years[19] also fit this trend very well. We point out that while the peak inclination anomalies in both the dynamo models and the PSV10 dataset tend to produce shallower than expected directions, the peak in the models is nearly always observed at latitudes of 25–30° (north or south; Appendix 1) whereas in the data it is within 10° of the equator[19]. If we nevertheless take the relationship in Fig. 4b at face value, the range of published Model G $a$ parameters from much older datasets suggest that its violations for the time periods studied here are unlikely to be much more severe than that measured for the last 10 Myr. A recent study[48] claimed that the model underlying the inclination anomalies measured for the past 10 Myr may be GAD; if this is true, then we cannot discount GAD for any of the periods examined here.

The overall picture emerging from this study is of a geomagnetic field whose average morphology has been extraordinarily uniformitarian in the face of substantial changes in geodynamo forcing that impacted on its strength and tendency to reverse polarity. It should be emphasised that this does not preclude the past occurrence of intervals of sustained highly anomalous field behaviour that also presented distinctive morphological characteristics (e.g. the mid-Palaeozoic[49,50] and Ediacaran[33,51] are both potential candidates for such times). It would, however, seem to require that such intervals are relatively rare and do not include the most recent superchron.

## Methods

**Calculation of virtual geomagnetic pole dispersion and Model G fits**. Outputs of magnetic field at Earth's surface were extracted from numerical dynamo and observational models in the form of 120 Gauss coefficients (i.e. up to degree and order 10) for each regularly spaced time realisation. We truncate the numerical dynamo simulation results to degree and order 10 in order to make them compatible with the highest resolution available in the observational models considered here.

In all analyses, except for those employing "down sampling" (Fig. 3b, c and Supplementary Fig. 4), 324 locations spaced 20° apart in longitude (between 0° and 340°) and 10° apart in latitude (between −85° and 85°) were analysed and 500 different sets of random timesteps were chosen at each of these. Note that this geographical sampling was deliberately chosen to be far from uniform (being very heavily concentrated at high latitudes) in order to define Model G equally well at all latitudes.

From each set of Gauss coefficients, we synthesised a magnetic field vector at the specified location and used its direction (expressed by declination and inclination) to represent an independent palaeomagnetic direction. Conversion to VGPs followed standard palaeomagnetic convention[52]. VGPs were then grouped by location, flipped to give a common polarity (i.e., the VGPs falling into the southern hemisphere were replaced by antipodal locations in the northern hemisphere), and used to produce 324 estimates of apparent palaeolatitude ($\lambda$) and VGP dispersion ($S$). $\lambda$ was calculated using the great-circle distance between the mean VGP position and the site location. $S$ was initially defined from the root mean square angular distances ($\Delta_i$) between the $i$th VGP and the mean VGP position according to

$$S = \left[ \frac{1}{N-1} \sum_{i=1}^{N} \Delta_i^2 \right]^{1/2}, \tag{6}$$

where $N$ is the total number of VGPs (500 in this case). This approach has been applied to all palaeomagnetic datasets used in our study, except for the 0–10 Ma dataset, and was therefore simulated here.

An iterative procedure was then used to exclude outliers at each location caused by reversal transitions and excursions following the well-established variable cut-off approach of Vandamme[29].

Model G (Eq. (4)) was fit to curves comprising the 324 $\lambda$–$S$ pairs calculated above using a least-squares minimisation algorithm within the optimisation toolbox of *Matlab* using a bounded search, where the limits are conservatively set for Model G $a$ and $b$ parameters (1–90° and 0–1, respectively). With the exception of the lower bound for $b$, all Model G fits fall far from the boundaries used in the minimisation.

The procedure for obtaining Model G parameters from down-sampled models was identical to the above except that $N$ at each location was reduced from 500 to 15 and the number of locations was reduced from 324 to 19 which were randomly drawn from a uniform distribution on a sphere.

**Numerical dynamo simulations**. Most of the numerical geodynamo models employed in this study have been extensively described elsewhere[53–57] and we thus outline only the essentials here. An electrically conducting and convecting Boussinesq fluid is confined in a spherical shell of thickness $d = r_o - r_i$, where $r_i$ and $r_o$ denote the inner and outer boundary radii, respectively. The spherical shell rotates about the vertical direction with angular frequency $\Omega$. As detailed in ref. [56], we solve numerically the momentum equation for the fluid velocity $\boldsymbol{u}$ in the co-rotating frame of reference, the induction equation for the magnetic field $\boldsymbol{B}$, and an equation of evolution for the temperature perturbations $T$. The equations are non-dimensionalised using the shell thickness $d$ as length scale, the core magnetic diffusion time $\tau_\eta = d^2/\eta$ as time scale, while $(2\Omega\rho\mu_0\eta)^{1/2}$ serves to rescale the magnetic field. Here $\eta$ denotes the outer core magnetic diffusivity, $\rho$ the core fluid density, and $\mu_0$ the vacuum permeability. Five dimensionless parameters control the system: the shell aspect ratio

$$\chi = \frac{r_i}{r_o}, \tag{7}$$

the Ekman number

$$E = \frac{\nu}{2\Omega d^2}, \tag{8}$$

the Prandtl number

$$Pr = \frac{\nu}{\kappa}, \tag{9}$$

the magnetic Prandtl number

$$Pm = \frac{\nu}{\eta}, \tag{10}$$

and the modified Rayleigh number

$$Ra = \frac{\alpha g_o \delta T d}{2\Omega\kappa}. \tag{11}$$

Here $\nu$, $\kappa$, and $\alpha$ are the fluid kinematic viscosity, thermal diffusivity, and thermal expansivity respectively; $g_o$ is gravity at the outer boundary and $\delta T$ is a temperature scale that depends on the temperature boundary conditions and on the internal heating mode (see refs. [53,55] for further details).

Supplementary Data 1 provides values of the above input parameters for all the numerical simulations employed in this study. All simulations have $Pr = 1$. With the exception of three models with smaller inner core sizes, we consider a present-day outer core aspect ratio of $\chi = 0.35$. All simulations employ no-slip flow boundary conditions and consider an electrically insulating mantle. The inner core can be either electrically insulating or conducting. As for the thermal boundary conditions, fixed heat flux (FF) is imposed at $r_o$ in all simulations. FF or fixed temperature (FT) conditions are used at $r_i$. Some simulations employ spatial variations in the outer boundary heat flux. In most of these cases, the imposed heat flux heterogeneity pattern is a recumbent spherical harmonic of degree 2 and order 0 (recumbent $Y_2^0$) that approximates the large-scale structure of the observed lower mantle seismic shear-wave anomalies[57,58]. Three models are instead based on the lower mantle tomographic model of shear-wave velocity of ref. [59]. The heterogeneity amplitude is defined by the parameter

$$\epsilon = \frac{q_{max} - q_{min}}{\langle q \rangle}, \tag{12}$$

where $q_{min}$ and $q_{max}$ are the minimum and maximum values of the outer boundary heat flux, respectively, and $\langle q \rangle$ is its surface mean value. Values of $\epsilon$ range from 0.3 to 1.5 in our numerical simulations (see Supplementary Data 1).

In the suite of simulations considered in this study, 37 have been reported in ref. [55] (a subset of these are previously published models; see Supplementary Table 1) and we thus do not describe them in detail here. We additionally employed 24 new simulations here. Among these, three include a uniform internal heat source term in the temperature equation modelling the presence of radiogenic heating (or secular cooling of the core). In several of these new models convection is purely chemically driven, that is the source of buoyancy is the release of light elements at the inner core boundary as the inner core freezes. Finally, some models allow for the presence of a stably stratified layer at the top of the core. We now briefly describe the formulation employed to model these different physical characteristics of the core. The equation of evolution for the temperature

perturbations $T$ around the background (adiabatic) reference state is

$$\frac{\partial T}{\partial t} + (\mathbf{u} \cdot \nabla)T = q\nabla^2 T + q\gamma. \quad (13)$$

Here $q = \kappa/\eta$ is the Roberts number, which is related to the input model parameters by $q = \mathrm{Pm}/\mathrm{Pr}$, and $\gamma$ is a uniform volumetric sink ($\gamma < 0$) or source ($\gamma > 0$) term. The stationary background temperature profile is given by

$$\frac{dT_0}{dr} = -\frac{\gamma}{3}r - \frac{1}{r^2} \quad (14)$$

with $\gamma = \gamma' d^2/\kappa\delta T$, where $\gamma'$ denotes the dimensional heat source/sink amplitude. A volumetric sink term and a zero heat flux condition at the outer boundary are appropriate for modelling purely chemical convection[60,61]. In this case, the variable $T$ here is interpreted as the relative concentration of light elements in the core that are released at the inner core boundary. From Eq. (M9), a zero flux condition at $r_o$ sets the value of the sink term to $\gamma = -3(1 - \chi)^3$. For $\chi = 0.35$, the present-day outer core aspect ratio, then $\gamma \approx -0.824 = \gamma_0$. For values $\gamma < \gamma_0$, the neutrally buoyant radius $r_*$ falls within the fluid interior. Convection thus occurs for $r < r_*$, while the region $r > r_*$ is sub-adiabatic and mimics the presence of a stably stratified layer at the top of the core. In our numerical simulations, we used either $\gamma = -1.14$ or $\gamma = -1.44$ (see Supplementary Data 1), which correspond to a stably stratified layer at the top of the core of thickness of about $\delta/d = 0.16$ and $\delta/d = 0.26$, respectively. In one case we explored the effect of an extreme stably stratified layer thickness of $\delta/d = 0.54$ ($\gamma = -3$).

Time is rescaled to physical units based on the electrical conductivity estimates provided by ref. [62] which suggests $\tau_\eta = 200$ kyr. All models were truncated such that transient effects associated with initialisation were excluded. The individual Gauss coefficients were then temporally resampled using a cubic spline fit in order to yield regularly spaced time steps.

**Regression and calculation of uncertainties.** Uncertainties for the estimates of AD/NAD$_\text{median}$ calculated for the palaeomagnetic datasets (Table 1, Fig. 4a) and the downsampled models (Fig. 3b, c) combined errors in the prediction of the power law (Fig. 2) and in the Model G $a$ parameter. The former were 95% prediction bounds on the power law displayed in Fig. 2 calculated using standard linear regression analysis and a t-distribution (Matlab curve-fitting toolbox and predint function using default settings) performed on the datasets in log-space. Although these techniques strictly assume Gaussian bivariate distributions, they are demonstrably effective here in encompassing the majority of the data. The latter consisted of 95% confidence bounds calculated using 1000 or 10,000 bootstraps resampling with replacement. Combining these two errors into a single uncertainty for estimated AD/NAD$_\text{median}$ allowed for the full overlap of error bars and the shaded region in Fig. 3c producing a conservative range whose usefulness is supported by the down-sampling results displayed in Fig. 3b, c and Supplementary Fig. 4.

## Data availability
The datasets generated during and/or analysed during the current study are available from the corresponding author on reasonable request.

## Code availability
The code used to perform these analyses are available from the corresponding author on reasonable request.

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

## Acknowledgements

A.J.B., C.J.S. and C.J.D. acknowledge support from the Natural Environment Research Council (standard grant, NE/P00170X/1); A.J.B., R.K.B. and D.G.M. acknowledge support from The Leverhulme Trust (Research Leadership Award, RL-2016-080); P.V.D. acknowledges support from the Research Council of Norway through its Centres of Excellence funding scheme, project 646 number 223272 (CEED).

## Author contributions

A.J.B. designed the study and performed the analyses. R.K.B., D.G.M., C.J.S. and C.J.D. performed new dynamo simulations and analyses. P.V.D. derived the theoretical power law. All above authors and R.H. contributed to writing the paper.

## Competing interests

The authors declare no competing interests.
