## [Peer Review File · Nature Communications]

Reviewers' comments:

Reviewer #1 (Remarks to the Author):

In this work the authors present an argument that the degree of axial dipole dominance in the geomagnetic field has remained essentially unchanged over timescales up to a billion years. This would certainly be re-assuring news for those relying on the geocentric axial dipole (GAD) hypothesis for plate reconstructions in the deep past. Given changes that might be expected associated with Earth's thermal evolution they also suggest that this presents a challenge for future core models.

However, I have some concerns with their analyses. I believe these preclude publication of this work, and will require the authors to undertake a major re-evaluation of all their results.

Dipole dominance is defined here by the ratio of the axial dipole term's contribution to that of the sum of other terms in Lowes' power spectrum evaluated at Earth's surface. Each term contributing to this spatial power spectrum evaluates the average squared field strength over Earth's surface, at the specified spherical harmonic degree n and order m . For modern field models and numerical dynamo simulations it is possible to use estimated gauss coefficients to evaluate AD/NAD directly and as a function of time, and likewise for those time varying paleomagnetic field models extending back to 100 ka constructed from both intensity and directional data. The quality of these estimates is of course wildly disparate because of limited accuracy and temporal and spatial coverage in the paleodata, and older times present an even greater challenge. Such estimates as do exist prior to 100 ka rely on Giant Gaussian Process models or on "Model G" as the authors propose in this manuscript.

The theory as provided in Model G parametrizes directional variability in the time-varying paleomagnetic field (specified via VGP dispersion about the geographic axis, S) in terms of contributions a and b with

$$S^2 = a^2 + (b\lambda)^2,$$

where a is the value at the equator, and has in the past been taken to represent variability contributed by spherical harmonics that are symmetric about the equator, and b the contributions from antisymmetric variations. The largest detectable contributions to b over the past few million years is variability in the axial dipole whose average dominance is the basis for the geocentric axial dipole hypothesis remaining the workhorse for reconstruction for paleo-latitudes in the deep past. The ratio b/a has been widely surmised to serve as a proxy for dominance of dipole power over other contributions, with large values of b

corresponding to dominance of equatorially antisymmetric terms, and g_1^0 being the dominant contributor to that variation. Thus b/a has been considered as a possible proxy for reversal rate over time, although this has been called into question by Doubrovine et al. (2019).

The goal of the current work is to test relationships between axial dipole dominance and a using results from a wide range of numerical dynamo simulations. It will be no surprise at all that high axial dipole dominance leads to low overall VGP dispersion at the equator and a steady increase with latitude. That result is largely confirmed here with numerical simulations.

The further challenge undertaken by the authors of the current paper is to translate average directional variability into a measure of AD/NAD over long time periods. To address this they present an empirical relationship between the log of pointwise median AD/NAD and $\log a$. It is unclear to me what the message from Figure 2 should be or whether it is useful. The reasons for my skepticism are related to the mish-mash of statistical signals and observation based models being tested. More details are given below.

1. **AD/NAD:** To measure dipole dominance the authors have chosen to use the median (over time) of point wise estimates of the ratio AD/NAD . Let's suppose that a GGP model provides a valid representation of the statistics of PSV. Then both AD and NAD in this scenario may be something like χ^2 distributed with 2 and M degrees of freedom (M depending on n_{nmax} in equation (2)). Given that both of these are required to be positive at any instant in time, this does not strike me as a robust estimate of dipole dominance. There is great potential for statistical bias in the estimates. In the event that GAD holds over the averaging time used for predictions in the deep past the ratio of the average values $\overline{AD}/\overline{NAD}$ will be expected to be very large, as the average non-dipole field will be small. The same will be true of the parameters $Median(AD)/Median(NAD)$ which would be a far more robust test than the highly variable point wise ratio used. In the event that GAD holds and the non-axial dipole field averages to zero, the ratio should be infinite reflecting complete dipole dominance. This indicates that Figure 2 and the model fits are far from the robust measure the authors claim. To be robust and enable a viable comparison the median values for AD and NAD need to be estimated independently over the same time scales as a before taking the ratio.

2. **Paleomodel limitations:** Aside from the issues raised in point (1) the differences in a values among the various paleofield models may not have as much to do with actual differences in field behavior, as to the assumptions and limitations of the modeling strategies used. Models 1,2, and 6 are likely too short to be representative, with 1 and 2 biased by recent high dipole field values (and limited time for variation), and model 6 covering the Laschamp excursion interval, which is all about large AD variation.

The AD/NAD value for CP88 is based on an AD value for the modern field, and its power spectrum at higher values largely determines the NAD as evaluated in the manuscript. But taking account of the time-averaged field only, the average $\overline{AD/NAD}$ would be $2\overline{g_1^0}^2/3\overline{g_2^0}^2$ or $\overline{AD/NAD}=185$. This is a far cry from the value of 4.4 used in Figure 2, but more like what one would expect for GAD to hold in the time average. This would also seem to be more appropriate for predicting averages over the long time intervals shown in Figure 4, and would lead to an entirely different model from the point-wise median values used in Figure 2.

It could be argued that CP88 does not fit the paleomagnetic observations of S , and should probably be ruled out in this comparison with Model G types, given the low variance in dipole contributions. These lead to a very low b values in Model G, although note that this is apparently mis-estimated in the current work (see below). CJ98 and TK03 remedy the defect of CP88 leading to more realistic estimate of latitude variation in S , but since the average NAD contributions are $\overline{g_2^0} = -1.5 \mu\text{T}$ and $0 \mu\text{T}$ respectively, they also have extremely large values of $\overline{AD/NAD}$ (266 and ∞) and seem quite divergent from the analysis conducted here.

(3) Despite a lot of computational effort put into the numerical simulations, the scaling results in Figure 1 seem self-evident. Aside from concerns expressed above about how it is estimated, higher median AD/NAD will inevitably decrease VGP dispersion overall, as the dominant axial dipole moves VGP directions closer to the geographic axis, and the lower values of a reflects this. A more intriguing result is the side note in Line 97-99 that the independence of equatorially symmetric and antisymmetric families is not supported by dynamo simulations. Should we believe this? It requires a valid estimate of O/E as for AD/NAD to investigate this properly. The current version of Supplementary Figure 2 cannot be trusted.

(4) There are two components to Model G, but only a is used here, representing equatorially symmetric contributions. If Model G is valid, the value of b representing the latitudinal trend in S is also an important measure of PSV, perhaps largely reflecting the relative amplitude of axial dipole variations as the dominant equatorially antisymmetric term. It's not discussed here, but the authors noted in their Dec 2019 AGU presentation that there was no correlation between AD/NAD and b in their exploratory analyses. This is puzzling because of course there is intrinsic covariance in the statistical estimates for a and b . It would be interesting and potentially informative to see a valid estimate of the equivalent of Fig 2 for b .

(5) I am deeply skeptical of empirical linear log-log relationships with no underlying theory.

Minor points

“increasingly poorly known” sends a mixed message in word usage. How about “Going back through time much less is known about the degree of dominance” ?

line 114-115: Note also that CP88 is not the only GGP to use a white spectrum at the CMB. TK03 has the same property (with a different balance of antisymmetric to symmetric terms from CP88), but not CJ98.

Figure 1 caption - if there is a shaded area around the red line for the simulations, I could not see it. The fit for LEDA001 doesn't look particularly good to me. Residuals would show strong correlations in latitude, suggestive of significant contributions from terms other than g_1^0 .

Supplementary Table 1: Estimates of a for CP88 and JC98 seem to be in conflict with the results in Figures 1 and 5 of Johnson & Constable (1999) which respectively show values of $\sim 15^\circ$ and 10° compared with 20.2° and 12.9° here. And CP88 does not conform to the quadratic model at all: eyeballing a value for b from JC's Fig 1 would give something close to zero, whereas the prediction from Supp Table 1 would suggest a VGP dispersion of 21.25° at 60° latitude, a clear discrepancy.

line 119: t-distribution and R^2 values rely on statistics being drawn from a bivariate Gaussian statistical process. Should use the non-parametric Spearman's rank correlation test here.

Reviewer #2 (Remarks to the Author):

- 1) It is clear after reading the methods section, but I think it would be worth clarifying at a round line 34-35 that you are considering the AD/NAD ratio at Earth's surface not the core mantle boundary.
- 2) At line 73: Did you remove from analysis the spin-up time for the dynamos i.e. the first half (or so) a diffusion time it takes dynamos to adjust to being restarted with new control parameters? I am confident from the second row of Figure 1 that you did, but I think it is worth clarifying.
- 3) Looking at Figure 1's lower left panels (the Model G fits for dynamos LEDA001 and LEDA030) it looks like the data would be better fit by a higher than second order polynomial. I'm not suggesting you do that. But it suggests to me that fitting the Model G b parameter makes an important impact to relationship described in equation 5. Because the VGP dispersion vs. latitude plots are more complicated for lower AD/NAD_{median} values, would you need a lot more paleomagnetic data from such a time period to confidently recover the AD/NAD? Suppose you only had VGP dispersion data from exactly at the equator, would you systematically underestimate AD/NAD? Is it correct, and important to clarify, that even though equation 5 is predicting a relationship between the average VGP dispersion at the equator and field dipolarity – in practice as paleomagnetists we still need VGP dispersion data from all latitudes in order to estimate the AD/NAD?
- 4) Lines 88-89: Is there a relationship with the b-parameter?
- 5) From lines 122-127, Figure 3a, and Sup Figure 3: I didn't follow what was being done here. How are the simulations being rescaled differently between the different rescaled models (small symbols)? Why does that show the power-law is robust rather than an artifact of how you choose to rescale?
- 6) Line 152: It would be useful to see (maybe in a supplementary figure) VGP dispersion vs. latitude plots for these five ancient time datasets. Was the same method for excluding outliers of Vandamme used on these datasets?
- 7) Line 274: Could you change the notation for the shell aspect ratio so that you do not use a to mean two different parameters?
- 8) The downsampling method seems like a useful estimate of how confidently we could recover paleomagnetically the AD/NAD_{median} vs. Model G a-parameter curve. What would it look like if you repeated that random downsampling a large number of times to estimate a different sort of confidence interval? I expect those bounds may be a lot larger than the 95% confidence bounds produced by bootstrap resampling the dynamo AD/NAD_{median}, Model G a-parameter pairs that went into the regression. Would that be a strong support for the approach of using dynamo simulations to make field predictions? Because using only limited paleomagnetic VGP data it would have been very unlikely to observe the same power-law relationship.

Typos:

Line 244: replace[s]  replaced

The commas after equations M2-M5 could be confused as primes.

Figure 3c x-axis label: Mode[*i*] G a parameter

Line 364: Missing Lowes from the reference

Generally, some Model G a parameters are in italics and some are not

Reviewer #3 (Remarks to the Author):

Review on "Quantitative estimates of geomagnetic axial dipole dominance in deep geological time", by Biggin et al.

Referee: Julien Aubert (you can disclose my name)

This is an interesting and very carefully executed study that demonstrates a simple relationship between one of the classical coefficients used in the analysis of the paleomagnetic secular variation in deep geological

time and the degree of dipole dominance of Earth's magnetic field. A such relationship may appear intuitive at first sight, and it has therefore long been suspected that there should exist one. However, the devil is in the details and things become much more complicated as one dwelves into these details (e.g. Hulot and Gallet 1996). For this reason, the findings presented here bring a refreshing and simple view on this long-standing problem, and it is a certainly a good thing to now also have a useful and quantitative handle on it. That way, the relationship turns out into one of the only means one can access to dipole dominance in the deep geological past, and I believe that the result is strong enough to justify publication in Nature Communications. The authors also rightfully point out that this relationship may be useful to reaffirm the relevance of the hypotheses that underly the paleogeographical reconstruction of continental drift. I therefore also believe that this study is a worthy addition to the more general literature, and that it is relevant to the broader readership of the journal.

I do have a few concerns and questions, listed below.

1) Given the simplicity of this new result, I believe that the authors may have suspected that a theory may be within reach to explain it with only a handful of Gauss coefficients (possibly even two, the axial and equatorial dipole coefficients), and have made attempts to search for this theory. If such is the case, I believe it would be worth reporting these efforts. If this is not the case, it would also be interesting to point out the difficulties that prevented a theoretical prediction of the result. Looking back into Hulot and Gallet 1996, the matter is certainly complex but the contrast with the simplicity of this new result is certainly striking enough to warrant a few words of explanation.

2) I strongly suspect that the strength set by the new AD/NAD constraint in the deep geological past on geodynamo modelling is largely oversold. Far from the challenge that is put forward in the abstract and the main text, I believe that achieving AD/NAD ratios consistent with Figure 4a is in fact almost a given in any dynamo simulation that reasonably accounts for the geodynamic state of the core in the deep past and presents a dominant axial dipole. To illustrate this, I have taken the cases of Landeau et al. 2017, PEPI (a paper that would possibly be worth citing in your study) and computed the median AD/NAD at the various stages of the paleoevolution scenarios presented in this paper. The results are below :

History 1 Age (Myr) median AD/NAD

0	24
330	37
540	32
700	34
1000	39

History 2 Age (Myr) median AD/NAD

0	20
330	37
540	30
700	24
1000	25

As can be seen, despite the models of Landeau et al. 2017 having been constructed without explicit need to fit a particular value of AD/NAD, they all fall right within (or in a single case very close to) the ranges reported in Fig. 4a. A notable exception is of course the bistable hemispherical state reported by Landeau et al. 2017 for History 1 at 700 Myr which has AD/NAD=0.1 (more on this at point 3 below).

I believe that the paper stands well enough on its feet without having to oversell the strength of the constraint for geodynamo modelling. The argument on the strong constraint posed by the new relationship should be dropped from the abstract and the main text. Rather, I think it would be important to point out that this constraint in fact strenghtens the success of the already existing geodynamo modelling efforts in accounting for the paleoevolution of the geodynamo. The above results can be used and cited as personal communication if the authors decide to use them in their revision.

3) The numerical dynamo dataset that has been gathered for the study is impressive and very carefully referenced. There is however a concern on the choice of aspect ratios, which is heavily biased towards present-day values (0.35). There are only three cases with lower aspect ratios, and none with a fully liquid outer core (absent inner core). The problem is that current estimates for the age of the inner core are within the range 500 Myr-1 Gyr, according to one of the coauthors' own work in ref. 57, which is also incidentally used to define the time rescaling of dynamo simulations (high core electrical conductivity hypothesis). It is therefore problematic that the dataset does not include a single simulation without an inner core, as would be appropriate for Earth prior to 500 Myr to 1 Gyr. This is also precisely the situation where exotic magnetic fields can arise, such as the hemispherical case of Landeau et al. 2017.

I think that the study would gain a lot from the addition of a few cases without an inner core and discussion of the possibility of exotic fields in this configuration, and how the new AD/NAD constraint provided by the authors could help to infirm/confirm this possibility.

In distant relationship to the above, perhaps I am not very up-to-date with the specific literature, but I am also a bit surprised that the line of work initiated by Abrajevitch and Van Der Woog 2010 (PEPI) on the possibility of an equatorial dipole-dominated geodynamo at Ediacaran epochs is also not discussed here.

4) The paper mostly focuses on the relationship between model G b and AD/NAD, but I believe that the lack of correlation between model G b/a and O/E is possibly an equally important result, that may put an end to some misinterpretations of model G. In the current paper state, I feel that this result is not being brought enough in the forefront. Maybe adding a sentence in the abstract and/or bringing back the supplementary figure among the main figures would better serve this purpose.

Julien Aubert

Response to reviewers

We are very grateful to all three reviewers for their useful comments and have performed substantial new analyses and revisions to address their concerns. These changes have included replacing one supplementary figure and making two entirely new ones, adding two sections of supplementary text, and making significant revisions to the main text. Since, our original manuscript was submitted, three new models of the magnetic field for the last 10 Myr based on the giant Gaussian Process have been published. We have therefore updated our analysis to include these – they all add further support to our conclusions.

Responses to Reviewer #1's comments

In this work the authors present an argument that the degree of axial dipole dominance in the geomagnetic field has remained essentially unchanged over timescales up to a billion years. This would certainly be re-assuring news for those relying on the geocentric axial dipole (GAD) hypothesis for plate reconstructions in the deep past. Given changes that might be expected associated with Earth's thermal evolution they also suggest that this presents a challenge for future core models.

However, I have some concerns with their analyses. I believe these preclude publication of this work, and will require the authors to undertake a major re-evaluation of all their results.

We thank the reviewer for their detailed comments and have indeed undertaken a substantial revision of the manuscript including several new analyses and figures. These changes have not significantly affected the findings of the study however which we contest were already correct and well-founded.

Dipole dominance is defined here by the ratio of the axial dipole term's contribution to that of the sum of other terms in Lowes' power spectrum evaluated at Earth's surface. Each term contributing to this spatial power spectrum evaluates the average squared field strength over Earth's surface, at the specified spherical harmonic degree n and order m . For modern field models and numerical dynamo simulations it is possible to use estimated gauss coefficients to evaluate $AD=NAD$ directly and as a function of time, and likewise for those time varying paleomagnetic field models extending back to 100 ka constructed from both intensity and directional data. The quality of these estimates is of course wildly disparate because of limited accuracy and temporal and spatial coverage in the paleodata, and older times present an even greater challenge. Such estimates as do exist prior to 100 ka rely on Giant Gaussian Process models or on "Model G" as the authors propose in this manuscript.

The theory as provided in Model G parametrizes directional variability in the time-varying paleomagnetic field (specified via VGP dispersion about the geographic axis, S) in terms of contributions a and b with

$$S^2 = a^2 + (b)^2;$$

where a is the value at the equator, and has in the past been taken to represent variability contributed by spherical harmonics that are symmetric about the equator, and b the contributions from antisymmetric variations. The largest detectable contributions to b over the past few million

years is variability in the axial dipole whose average dominance is the basis for the geocentric axial dipole hypothesis remaining the workhorse for reconstruction for paleo-latitudes in the deep past. The ratio b/a has been widely surmised to serve as a proxy for dominance of dipole power over other contributions, with large values of b corresponding to dominance of equatorially antisymmetric terms, and g_{01} being the dominant contributor to that variation. Thus b/a has been considered as a possible proxy for reversal rate over time, although this has been called into question by Dubrovine et al. (2019).

The goal of the current work is to test relationships between axial dipole dominance and a using results from a wide range of numerical dynamo simulations. It will be no surprise at all that high axial dipole dominance leads to low overall VGP dispersion at the equator and a steady increase with latitude. That result is largely confirmed here with numerical simulations.

We agree with the reviewer on every point made above but would like to clarify that, although we also consider the qualitative result to be unsurprising, we did not only confirm this but rather put it on a quantitative footing which we argue, and the other reviewers agree, is the transformative finding here. This is made clear in the title, abstract and main text. We have since supplemented this finding with a complementary theoretical treatment in the supplement, discussed below.

The further challenge undertaken by the authors of the current paper is to translate average directional variability into a measure of $AD=NAD$ over long time periods. To address this they present an empirical relationship between the log of pointwise median $AD=NAD$ and $\log a$. It is unclear to me what the message from Figure 2 should be or whether it is useful. The reasons for my skepticism are related to the mish-mash of statistical signals and observation based models being tested. More details are given below.

We have now added clarification to the text to try to ensure that doubt is removed over the key message in Figure 2 and its high degree of usefulness. Details are given below.

1. AD/NAD : To measure dipole dominance the authors have chosen to use the median (over time) of point wise estimates of the ratio AD/NAD . Let's suppose that a GGP model provides a valid representation of the statistics of PSV. Then both AD and NAD in this scenario may be something like χ^2 distributed with 2 and M degrees of freedom (M depending on n_{max} in equation (2)). Given that both of these are required to be positive at any instant in time, this does not strike me as a robust estimate of dipole dominance. There is great potential for statistical bias in the estimates. In the event that GAD holds over the averaging time used for predictions in the deep past the ratio of the average values $AD=NAD$ will be expected to be very large, as the average non-dipole field will be small. The same will be true of the parameters $Median(AD)=Median(NAD)$ which would be a far more robust test than the highly variable point wise ratio used. In the event that GAD holds and thenon-axial dipole field averages to zero, the ratio should be infinite reflecting complete dipole dominance. This indicates that Figure 2 and the model fits are far from the robust measure the authors claim. To be robust and enable a viable comparison the median values for AD and NAD need to be estimated independently over the same time scales as a before taking the ratio.

We agree with the reviewer that both AD and NAD are required to be positive at any one point in time but do not share the view that this is problematic and are a bit confused over the biasing referred to. Perhaps the confusion occurs because the reviewer is considering the classical time-averaged field whereas we are only seeking the average of the instantaneous power ratios? To improve clarity in the manuscript, we have made two new supplementary figures (Supp figs 2 and 3)

which clearly compare the same results obtained using time-instantaneous medians and the classical time-average field. We have also updated Supplementary Tables 1 and 2 with this information.

Time-average field (TAF) results obtained by time-averaging the power independently is a more useful property of the field for the classical applied palaeomagnetist (e.g. engaged in tectonic reconstructions etc) to know. However, the inclination anomaly which we already calculate (and plot versus Model G a in Figure 4b) is more useful still for such scientists. Rather, in Figure 2, we are seeking what we consider to be a useful property for the palaeo-geomagnetist, dynamo modeller and palaeo-magnetosphericist –i.e. the time-averaged axial dipole dominance rather than the axial dipole dominance of the time-averaged field. We have therefore adapted the manuscript to make it clear what we are seeking and why (Lines 40-50). **“We note that this value is a first-order description of the average, time-instantaneous field morphology and is not intended as a direct measure of the validity of the geocentric axial dipole (GAD) hypothesis which rather would rely on the morphology of the time-average field (TAF). The TAF field is defined by time-averaging all Gauss coefficients independently before using their ratios to define its properties, which may be very different to the properties of the instantaneous field at any and all times. For example AD/NAD_{TAF} is, by definition, infinite for a GAD field whereas the associated AD/NAD_{median} value may be finite and even small. In this sense, AD/NAD_{median} is more relevant to those using palaeomagnetic records to understand geomagnetic behaviour, core dynamics and the magnetospheric shielding it confers than to those interested in making tectonic reconstructions. The implications of this study for palaeogeographical reconstructions is nevertheless explored later.”**

We tested the reviewer’s assertion that $Median(AD)/Median(NAD)$ would go to infinity in a GAD field using 10,000 realisations of TK03.GAD but found it to be incorrect (see figure below for individual AD and NAD values). In fact it gives a value (12.8) rather similar to that of $Median(AD/NAD)$ (12.2). We also consider it to be less intuitive (since our parameter effectively defines the most common state of the field) so prefer to stay with our original parameter. In order for AD/NAD to go towards infinity, it is required that NAD values are calculated from time averages of individual gauss coefficients that are then used to describe a time-averaged NAD (rather than an average power of many NAD fields). This is a ratio we now refer to as AD/NAD_{TAF} and give in Supplementary Table 1; it is indeed very high.

Figure R1: AD and NAD values obtained from 10,000 realisations of TK03.GAD

2. Paleomodel limitations: Aside from the issues raised in point (1) the differences in a values among the various paleofield models may not have as much to do with actual differences in field behavior, as to the assumptions and limitations of the modeling strategies used. Models 1,2, and 6 are likely too short to be representative, with 1 and 2 biased by recent high dipole field values (and limited time for variation), and model 6 covering the Laschamp excursion interval, which is all about large AD variation.

We do not contest this point but rather highlight that it is not of primary relevance to the purpose that these models serve in this study (to validate our proxy). Whether the differences are strictly relevant to Earth or not, they exist in the models and the associated proxy (also derived from the models) tracks them very well. We suspect this to be true for any model whose power spectrum is not wildly different from Earth. We have now added the following text at lines 144-147 to make this clear: **“We note that we are not overly concerned here with the relative realism of any of the outputs shown by these models, merely the ability of their output palaeosecular variation to predict their average morphology.”**

The $\overline{AD}/\overline{NAD}$ value for CP88 is based on an AD value for the modern field, and its power spectrum at higher values largely determines the NAD as evaluated in the manuscript. But taking account of the time-averaged field only, the average $\overline{AD}/\overline{NAD}$ would be $2g_1^0 / 3g_2^0$ or $\overline{AD}/\overline{NAD} = 185$. This is a far cry from the value of 4.4 used in Figure 2, but more like what one would expect for GAD to hold in the time average. This would also seem to be more appropriate for predicting averages over the long time intervals shown in Figure 4, and would lead to an entirely different model from the point-wise median values used in Figure 2.

The value of 4.4 was incorrect and due to an error in our code (see below). The correct value now calculated is 13.4. This is still far from 185 and this discrepancy is again due to the point referred to above regarding what we are trying to achieve in Figure 2. $\overline{AD}/\overline{NAD} = 185$ is a conventional time average field description, useful for applied palaeomagnetists (though not as useful as the inclination anomaly which we calculate later) but not useful in characterising the instantaneous field morphology. The latter is what we are seeking to obtain by proxy (Model G *a*) in Figure 2 and has a time-average value of 13.4. A value of 188.5 (marginally higher than 185 because the 10,000 iterations were not sufficient to average all other terms to zero) is found (and shown in the revised Supplementary Table 1 and new Supplementary Figure 3e,f) where we investigate the time-average field, with a ratio we refer to as AD/NAD_{TAF} .

It could be argued that CP88 does not fit the paleomagnetic observations of S, and should probably be ruled out in this comparison with Model G types, given the low variance in dipole contributions.

We agree with the first part of this sentence but we do not think that this precludes its usefulness here as a test of the proxy (see response above). It is not implausible that low variance from dipolar contributions was a characteristic of the palaeo-field at some point in the past. If it were, Model G *a* is demonstrated to be a useful proxy to tell us this.

These lead to a very low b values in Model G, although note that this is apparently mis-estimated in the current work (see below).

CJ98 and TK03 remedy the defect of CP88 leading to more realistic estimate of latitude variation in S, but since the average NAD contributions are $g_{02} = -1.5$ T and 0 T respectively, they also have extremely large values of AD/NAD (266 and ∞) and seem quite divergent from the analysis conducted here.

We refer to the answer given above. The values of 266 and infinity are only true of the conventional time average field, not the average instantaneous axial dipole dominance which is what we are seeking. Again, we now show AD/NAD_{TAF} values similar values to those mentioned in Supplementary Table 1 and Supplementary Figure 3e,f.

Despite a lot of computational effort put into the numerical simulations, the scaling results in Figure 1 seem self-evident. Aside from concerns expressed above about how it is estimated, higher median $AD=NAD$ will inevitably decrease VGP dispersion overall, as the dominant axial dipole moves VGP directions closer to the geographic axis, and the lower values of a reflects this.

We agree that the qualitative result is expected and already acknowledge this (Lines 118-121) alongside the substantial and unexpected finding, regarding the quantitative power of the proxy across a huge range of theoretical and observational models.

A more intriguing result is the side note in Line 97-99 that the independence of equatorially symmetric and antisymmetric families is not supported by dynamo simulations. Should we believe this? It requires a valid estimate of O/E as for AD/NAD to investigate this properly. The current version of Supplementary Figure 2 cannot be trusted.

We have replaced the original Supplementary Figure 2 with two new multi-panel figures (Supplementary Figures 2 and 3). In the previous version, we included g_{10} as an odd term. This has been changed here to make consistent with Coe & Glatzmaier (2006). The result is shown in panels a and d of Supplementary Figure 3. There is some positive covariance but it is somewhat weak for both the median instantaneous fields and the TAF. We have therefore modified the text at lines 108-116 to read: **“We also observe the following: (1) since Model G a and b parameters co-vary**

(Supplementary Figure 2a), the latter is also correlated with AD/NADmedian but here the relationship is not quite so strong (Supplementary Figure 3c); (2) the relatively weak relationship between b/a and O/E implies that the original morphological interpretation of Model G parameters in terms of independent families of equatorially symmetric and antisymmetric spherical harmonic terms 5, 18, 22 is only moderately supported by our dynamo simulations (Supplementary Figure 3a,d); (3) intuitively, Model G parameters provide much stronger constraints on the average instantaneous field morphology (Supplementary Figure 3a,b,c) than the morphology of the time-averaged field (Supplementary Figure 3d,e,f)."

There are two components to Model G, but only a is used here, representing equatorially symmetric contributions. If Model G is valid, the value of b representing the latitudinal trend in S is also an important measure of PSV, perhaps largely reflecting the relative amplitude of axial dipole variations as the dominant equatorially antisymmetric term.

It's not discussed here, but the authors noted in their Dec 2019 AGU presentation that there was no correlation between AD=NAD and b in their exploratory analyses. This is puzzling because of course there is intrinsic covariance in the statistical estimates for a and b. It would be interesting and potentially informative to see a valid estimate of the equivalent of Fig 2 for b.

This would indeed be puzzling so we think we must have been unclear in what we said at AGU2019. The reviewer is correct that in most of the dynamo models, b co-varies with a and both a and b co-vary with AD/NAD. We now show this explicitly in the new Supplementary Figures 2 and 3. The parameter a has a stronger correlation with AD/NAD than does b and therefore we use this as our proxy.

(5) I am deeply skeptical of empirical linear log-log relationships with no underlying theory.

We are a bit confused by this comment but, in any case, have now added a section of Supplementary Text in which we theoretically derive a power law with similar constants to that obtained from the empirical fit to the models' outputs. This is referred to in the text at lines 121-122.

Minor points

"increasingly poorly known" sends a mixed message in word usage. How about "Going back through time much less is known about the degree of dominance"?

Corrected as requested

line 114-115: Note also that CP88 is not the only GGP to use a white spectrum at the CMB. TK03 has the same property (with a different balance of antisymmetric to symmetric terms from CP88), but not CJ98.

This sentence has been updated in light of this comment.

Figure 1 caption - if there is a shaded area around the red line for the simulations, I could not see it. The fit for LEDA001 doesn't look particularly good to me.

Residuals would show strong correlations in latitude, suggestive of significant contributions from terms other than g01.

We have updated the caption to make it clear that uncertainties only apply to the observational dataset (since they could be made arbitrarily small by resampling in the case of simulations). We do

not disagree with the other comments but point out that the fit is sufficiently good to allow useful predictions to be made.

Supplementary Table 1: Estimates of a for CP88 and JC98 seem to be in conflict with the results in Figures 1 and 5 of Johnson & Constable (1999) which respectively show values of 15 and 10 compared with 20:2 and 12:9 here.

We presume this refers to Constable & Johnson (1999)? If so, we are very grateful to the reviewer for spotting this discrepancy as it was caused by an error in our code used to generate sets of gauss coefficients from CP88 and CJ98. We have now corrected the code and found much better agreement. Minor discrepancies with the original do exist but are likely caused by second-order differences in precisely how the values were calculated (which was not entirely clear in the original version of CJ98).

And CP88 does not conform to the quadratic model at all: eyeballing a value for b from JC's Fig 1 would give something close to zero, whereas the prediction from Supp Table 1 would suggest a VGP dispersion of 21.25 at 60 latitude, a clear discrepancy.

We agree. This was due to the same coding error which has now been fixed giving a considerably lower value of b (as well as a). The figure (left) can be compared directly to Fig1 in Constable and Johnson (1999).

. line 119: t-distribution and R2 values rely on statistics being drawn from a bivariate Gaussian statistical process. Should use the non-parametric Spearman's rank correlation test here.

We now acknowledge that this assumption is violated (lines 384-385) and provide the Spearman's rank correlation ($\rho = -0.971$) with the R^2 at line 120. We nevertheless retain the prediction bounds because we are not aware of a good nonparametric means of obtaining them and the existing ones certainly appear to do a good job.

Response to Reviewer #2's comments

1) It is clear after reading the methods section, but I think it would be worth clarifying at around line 34-35 that you are considering the AD/NAD ratio at Earth's surface not the core mantle boundary.

Mentioned at Line 26

2) At line 73: Did you remove from analysis the spin-up time for the dynamos i.e. the first half (or so) a diffusion time it takes dynamos to adjust to being restarted with new control parameters? I am confident from the second row of Figure 1 that you did, but I think it is worth clarifying.

Yes, up to the first diffusion time was removed to ensure any transient had passed. This has now been noted in lines 376-377.

3) Looking at Figure 1's lower left panels (the Model G fits for dynamos LEDA001 and LEDA030) it looks like the data would be better fit by a higher than second order polynomial. I'm not suggesting you do that. But it suggests to me that fitting the Model G b parameter makes an important impact to relationship described in equation 5. Because the VGP dispersion vs. latitude plots are more complicated for lower AD/NAD_median values, would you need a lot more paleomagnetic data from such a time period to confidently recover the AD/NAD?

Yes, it seems to stand to reason that the number of data and particularly their latitudinal distribution needs to be larger for at least the dynamo models with lower AD/NAD values - This has now been noted in lines 95-99.

Suppose you only had VGP dispersion data from exactly at the equator, would you systematically underestimate AD/NAD? Is it correct, and important to clarify, that even though equation 5 is predicting a relationship between the average VGP dispersion at the equator and field dipolarity – in practice as paleomagnetists we still need VGP dispersion data from all latitudes in order to estimate the AD/NAD?

This is a very good question. Yes, it seems you would for many of the models as this equatorial "peak" is seen repeatedly. We have also pointed this out in lines 95-99

4) Lines 88-89: Is there a relationship with the b-parameter?

Yes, and this has been addressed in the new Supp Fig 3 and the text at lines 108-110.

5) From lines 122-127, Figure 3a, and Sup Figure 3: I didn't follow what was being done here. How are the simulations being rescaled differently between the different rescaled models (small symbols)? Why does that show the power-law is robust rather than an artifact of how you choose to rescale?

To resolve this confusion, we have added a step-by-step description of the process workflow in the Supplementary text and a sentence to the main text – (Lines 154-156) **"This demonstrates that, so long as the power spectrum of the nondipole field is consistent with any of these models, the relationship is robust to a large range of AD/NADmedian values."**

6) Line 152: It would be useful to see (maybe in a supplementary figure) VGP dispersion vs. latitude plots for these five ancient time datasets. Was the same method for excluding outliers of Vandamme used on these datasets?

We now clarify that the Model G parameters were taken directly from the publications and also that, in each case, the Vandamme cutoff was applied (lines 184-185). The relevant figures are already published in the literature and we would prefer not to try to reproduce them.

7) Line 274: Could you change the notation for the shell aspect ratio so that you do not use a to mean two different parameters?

Done, thanks for pointing out.

8) The downsampling method seems like a useful estimate of how confidently we could recover paleomagnetically the AD/NAD_median vs. Model G a-parameter curve. What would it look like if you repeated that random downsampling a large number of times to estimate a different sort of confidence interval? I expect those bounds may be a lot larger than the 95% confidence bounds produced by bootstrap resampling the dynamo AD/NAD_median, Model G a-parameter pairs that went into the regression. Would that be a strong support for the approach of using dynamo simulations to make field predictions? Because using only limited paleomagnetic VGP data it would have been very unlikely to observe the same power-law relationship.

We have performed the requested analysis – a Monte Carlo simulation of 1000 downsamples of each of the models in figure 3c using the 25 and 975 ranked Model G a values as uncertainties. The results are shown in Supp Figure 6 and demonstrate that the 95% bounds are smaller than expected consistent with Figure 3c showing that individual downsample error bars tend to intercept the prediction bounds.

Typos:

Line 244: replace[s]  replaced

Done

The commas after equations M2-M5 could be confused as primes.

Fixed

Figure 3c x-axis label: Mode[l] G a parameter

Fixed

Line 364: Missing Lowes from the reference

Fixed

Generally, some Model G a parameters are in italics and some are not

Fixed

Response to Reviewer #3's comments

Review on "Quantitative estimates of geomagnetic axial dipole dominance in deep geological time", by Biggin et al.

Referee: Julien Aubert (you can disclose my name)

This is an interesting and very carefully executed study that demonstrates a simple relationship between one of the classical coefficients used in the analysis of the paleomagnetic secular variation in deep geological time and the degree of dipole dominance of Earth's magnetic field. A such relationship may appear intuitive at first sight, and it has therefore long been suspected that there should exist one. However, the devil is in the details and things become much more complicated as one dwelves into these details (e.g. Hulot and Gallet 1996). For this reason, the findings presented here bring a refreshing and simple view on this long-standing problem, and it is a certainly a good thing to now also have a useful and quantitative handle on it. That way, the relationship turns out into one of the only means one can access to dipole dominance in the deep geological past, and I believe that the result is strong enough to justify publication in Nature Communications. The authors also rightfully point out that this relationship may be useful to reaffirm the relevance of the hypotheses that underly the paleogeographical reconstruction of continental drift. I therefore also believe that this study is a worthy addition to the more general literature, and that it is relevant to the broader readership of the journal.

I do have a few concerns and questions, listed below.

1) Given the simplicity of this new result, I believe that the authors may have suspected that a theory may be within reach to explain it with only a handful of Gauss coefficients (possibly even two, the axial and equatorial dipole coefficients), and have made attempts to search for this theory. If such is the case, I believe it would be worth reporting these efforts. If this is not the case, it would also be interesting to point out the difficulties that prevented a theoretical prediction of the result. Looking back into Hulot and Gallet 1996, the matter is certainly complex but the contrast with the simplicity of this new result is certainly striking enough to warrant a few words of explanation.

We thank Julien for this suggestion. We have now added a section of Supplementary Text in which we theoretically derive a power law with similar constants to that obtained from the empirical fit to the models' outputs. This is referred to in the text at lines 121-122.

2) I strongly suspect that the strength set by the new AD/NAD constraint in the deep geological past on geodynamo modelling is largely oversold. Far from the challenge that is put forward in the abstract and the main text, I believe that achieving AD/NAD ratios consistent with Figure 4a is in fact almost a given in any dynamo simulation that reasonably accounts for the geodynamic state of the core in the deep past and presents a dominant axial dipole. To illustrate this, I have taken the cases of Landeau et al. 2017, PEPI (a paper that would possibly be worth citing in your study) and computed the median AD/NAD at the various stages of the paleoevolution scenarios presented in this paper. The results are below:

History 1 Age (Myr) median AD/NAD

0 24
330 37
540 32
700 34
1000 39

History 2 Age (Myr) median AD/NAD

0 20
330 37
540 30
700 24
1000 25

As can be seen, despite the models of Landeau et al. 2017 having been constructed without explicit need to fit a particular value of AD/NAD, they all fall right within (or in a single case very close to) the ranges reported in Fig. 4a. A notable exception is of course the bistable hemispherical state reported by Landeau et al. 2017 for History 1 at 700 Myr which has AD/NAD=0.1 (more on this at point 3 below).

I believe that the paper stands well enough on its feet without having to oversell the strength of the constraint for geodynamo modelling. The argument on the strong constraint posed by the new relationship should be dropped from the abstract and the main text. Rather, I think it would be important to point out that this constraint in fact strenghtens the success of the already existing geodynamo modelling efforts in accounting for the paleoevolution of the geodynamo. The above results can be used and cited as personal communication if the authors decide to use them in their revision.

We thank Julien for this insightful and helpful comment and have accepted his offer to cite this observation as a pers.comms (lines 224-226). We have similarly toned down the constraint arguments in the abstract and main text.

3) The numerical dynamo dataset that has been gathered for the study is impressive and very carefully referenced. There is however a concern on the choice of aspect ratios, which is heavily biased towards present-day values (0.35). There are only three cases with lower aspect ratios, and none with a fully liquid outer core (absent inner core). The problem is that current estimates for the age of the inner core are within the range 500 Myr-1 Gyr, according to one of the coauthors' own work in ref. 57, which is also incidentally used to define the time rescaling of dynamo simulations (high core electrical conductivity hypothesis). It is therefore problematic that the dataset does not include a single simulation without an inner core, as would be appropriate for Earth prior to 500 Myr to 1 Gyr. This is also precisely the situation where exotic magnetic fields can arise, such as the hemispherical case of Landeau et al. 2017.

I think that the study would gain a lot from the addition of a few cases without an inner core and discussion of the possibility of exotic fields in this configuration, and how the new AD/NAD constraint provided by the authors could help to infirm/confirm this possibility.

In distant relationship to the above, perhaps I am not very up-to-date with the specific literature, but I am also a bit surprised that the line of work initiated by Abrajevitch and Van Der Woog 2010 (PEPI) on the possibility of an equatorial dipole-dominated geodynamo at Ediacaran epochs is also not discussed here.

We have now caveated our findings with the acknowledgement that extreme multipolar or exotic field morphologies such as a persistent equatorial dipole could violate the relationship given here and that such situations may have arisen in the past (lines 228-233). We have not performed any simulations without an inner core as the code we are using does not allow it but have provided a few with smaller inner core sizes.

4) The paper mostly focuses on the relationship between model G b and AD/NAD, but I believe that the lack of correlation between model G b/a and O/E is possibly an equally important result, that may put an end to some misinterpretations of model G. In the current paper state, I feel that this result is not being brought enough in the forefront. Maybe adding a sentence in the abstract and/or bringing back the supplementary figure among the main figures would better serve this purpose.

In fact, a moderate-weak correlation does emerge when the axial dipole is dropped from the odd terms and this is newly shown in Supp. Fig 3a and referenced in the main text 110-114.

Julien Aubert

REVIEWER COMMENTS

Reviewer #1 (Remarks to the Author):

The authors have made revisions in response to the reviews that result in significant clarifications. I am glad to have been of assistance in helping to uncover their programming error, and offer my apologies for confusing the author order in the reference which they correctly surmise should have been to Constable & Johnson (1999).

The authors are fully convinced of the significance of their new result and refer to it as a quantitative proxy for estimating the dominance of g_0 over time. I hope that they prove correct. The essential result drawn from Figure 4 is that there is no discernible change in axial dipole dominance over time. I do worry a bit about the statistical robustness inherent in using the median of the ratio of 2 quadratic measures of the field here (AD/NAD in the author's terminology). This also means that the estimated bounds are wide. Would the results of Figures 2 and 4 hold equally if $VAD/VNAD_{median}$ were used instead? Presumably a log scale would no longer be required in that case? And perhaps the bounds recovered in Figure 4 (are they 95% level) could be narrower. But I suspect the authors have recovered what they wanted via the ir specific definitions.

The theoretical argument presented in the supplement supposes that all terms apart from g_0 average to zero, considers only variations in direction and strength of the overall dipole, and removes any latitudinal dependence setting $b = 0$. I'm not sure of the usefulness of this result, but I guess it does no harm, unless it is subsequently adopted as representative of the true geomagnetic field.

Reviewer #3 (Remarks to the Author):

I thank the authors for their detailed and relevant response to all of my comments. I think it is an interesting paper which in its revised form now gets my recommendation for acceptance in Nature communications.

Julien Aubert

Response to Reviewers' comments

Reviewer 1

The authors have made revisions in response to the reviews that result in significant clarifications. I am glad to have been of assistance in helping to uncover their programming error, and offer my apologies for confusing the author order in the reference which they correctly surmise should have been to Constable & Johnson (1999).

We thank reviewer 1 for their further comments and are very pleased that our revisions have clarified the main areas that they previously viewed as problematic.

The authors are fully convinced of the significance of their new result and refer to it as a quantitative proxy for estimating the dominance of g_{10} over time. I hope that they prove correct. The essential result drawn from figure 4 is that there is no discernible change in axial dipole dominance over time. I do worry a bit about the statistical robustness inherent in using the median of the ratio of 2 quadratic measures of the field here (AD/NAD in the author's terminology). This also means that the estimated bounds are wide. Would the results of Figures 2 and 4 hold equally if $\sqrt{AD}/\sqrt{NAD}_{median}$ were used instead? Presumably a log scale would no longer be required in that case? And perhaps the bounds recovered in Figure 4 (are they 95% level) could be narrower. But I suspect the authors have recovered what they wanted via their specific definitions.

We investigated the suggested parameter $\sqrt{AD}/\sqrt{NAD}_{median}$ but concluded that this provided no more information than AD/NAD_{median} since $\text{median}(\text{sqrt}(AD)/\text{sqrt}(NAD)) = \text{sqrt}(\text{median}(AD/NAD))$. A power law remains (see figures) and therefore a log axis for figure 4 is the most appropriate. We have clarified that the bounds in Figure 4 are 95% uncertainty limits in the main text (line 199) and figure 4 caption.

The theoretical argument presented in the supplement supposes that all terms apart from g_{10} average to zero, considers only variations in direction and strength of the overall dipole, and removes any latitudinal dependence setting $b = 0$. I'm not sure of the usefulness of this result, but I guess it does no harm, unless it is subsequently adopted as representative of the true geomagnetic field.

We certainly did not intend the theoretical derivation to be adopted as representative of the true geomagnetic field for reasons given. We have made this plainer in the supplementary text.

Reviewer 3

I thank the authors for their detailed and relevant response to all of my comments. I think it is an interesting paper which in its revised form now gets my recommendation for acceptance in Nature communications.

Julien Aubert

We are very grateful to Julien for his earlier comments and positive recommendation.

REVIEWERS' COMMENTS

Reviewer #1 (Remarks to the Author):

Thank you for the additional clarifications. I have no further comments on this manuscript. It seems ready for publication.